# The Application of Z-Numbers in Fuzzy Decision Making: The State of the Art

Nik Muhammad Farhan Hakim Nik Badrul Alam [1,2], Ku Muhammad Naim Ku Khalif [1,3,*], Nor Izzati Jaini [1] and Alexander Gegov [4,5,*]

1. Centre for Mathematical Sciences, Universiti Malaysia Pahang, Kuantan 26300, Pahang, Malaysia; farhanhakim@uitm.edu.my (N.M.F.H.N.B.A.); ati@ump.edu.my (N.I.J.)
2. Mathematical Sciences Studies, College of Computing, Informatics and Mathematics, Universiti Teknologi MARA Pahang Branch, Jengka 26400, Pahang, Malaysia
3. Centre of Excellence for Artificial Intelligence & Data Science, Universiti Malaysia Pahang, Kuantan 26300, Pahang, Malaysia
4. School of Computing, University of Portsmouth, Portsmouth PO1 3HE, UK
5. English Faculty of Engineering, Technical University of Sofia, 1756 Sofia, Bulgaria
* Correspondence: kunaim@ump.edu.my (K.M.N.K.K.); alexander.gegov@port.ac.uk (A.G.)

**Abstract:** A Z-number is very powerful in describing imperfect information, in which fuzzy numbers are paired such that the partially reliable information is properly processed. During a decision-making process, human beings always use natural language to describe their preferences, and the decision information is usually imprecise and partially reliable. The nature of the Z-number, which is composed of the restriction and reliability components, has made it a powerful tool for depicting certain decision information. Its strengths and advantages have attracted many researchers worldwide to further study and extend its theory and applications. The current research trend on Z-numbers has shown an increasing interest among researchers in the fuzzy set theory, especially its application to decision making. This paper reviews the application of Z-numbers in decision making, in which previous decision-making models based on Z-numbers are analyzed to identify their strengths and contributions. The decision making based on Z-numbers improves the reliability of the decision information and makes it more meaningful. Another scope that is closely related to decision making, namely, the ranking of Z-numbers, is also reviewed. Then, the evaluative analysis of the Z-numbers is conducted to evaluate the performance of Z-numbers in decision making. Future directions and recommendations on the applications of Z-numbers in decision making are provided at the end of this review.

**Keywords:** Z-number; fuzzy decision making; SWOT; fuzzy ranking; multi-criteria decision making

## 1. Introduction

Natural language (NL) is more understandable when a decision on certain things should be made as human beings use NL to describe almost everything in daily life. It is not accurate to describe things using crisp numbers 1 and 0 to represent the truth and falsity, respectively. For example, when a person is required to describe the weather of the day, then they are limited to describe the weather as 'rainy' or 'not rainy', in which the linguistic terms, if mathematically written, are the crisp numbers 1 or 0, respectively. Using the knowledge of fuzzy sets [1], the person can better describe the weather as 'very rainy', 'rainy', 'slightly rainy', 'cloudy', 'slightly cloudy', 'sunny', 'very sunny', etc. The application of fuzzy sets allows human beings to better describe their opinion in NL, and some uncertainties can be reduced.

Using the classical fuzzy sets, each element in the set is assigned a membership function in the interval [1] to depict its degree of belongingness to the set. Type-2 fuzzy set was then introduced by [2] to describe the membership function using a classical fuzzy

set (Type-1 fuzzy set) instead of any crisp value from the interval [1]. Hence, the Type-2 fuzzy set has a three-dimensional membership function. Realizing that it is not enough to represent the elements in the fuzzy sets by the membership grade, Atanassov [3] defined the non-membership grade that measures how much the elements do not belong to the fuzzy set. The hesitancy grade was also defined to complement the value between the membership and non-membership grades. In 2010, Torra [4] proposed a new concept of a fuzzy set called the hesitant fuzzy set, which allowed the element of the fuzzy sets to have more than one membership function. When it comes to decision making, the decision-makers have some hesitancy in describing the variable using a single membership grade. Hence, the hesitant fuzzy set allows the decision-makers to assign multiple membership grades for such variables. In an intuitionistic fuzzy set, the membership, non-membership, and hesitancy grades are treated as dependent variables. In contrast, Wang et al. [5] extended Smarandache's [6] work in defining the single-valued neutrosophic sets, in which the truth-membership, falsity-membership, and indeterminacy-membership are all treated as independent variables.

In the real world, decision information is almost always imperfect [7]. Information is imperfect when it is imprecise, incomplete, unreliable, and vague [8]. The imprecision can be described as either more than one or no realization match the available information [9]. According to [10], imperfect information leads to difficulties in giving accurate preferences among decision-makers. The decision information is not completely reliable due to the incompetency of decision-makers, complicated alternatives, and psychological biases [10].

According to Liu et al. [11], the traditional fuzzy numbers only depict nonlinear and uncertain information, in which the level of sureness of such information is ignored. Hence, Z-number was introduced by [12] to properly cater for imperfect information. The Z-number is composed of a pair of fuzzy numbers, (*A*, *B*). The first component, *A*, describes the restriction on the values that a random variable can take, while the second component, *B*, is the measure of reliability or certainty of *A*. Using the reliability component, the Z-numbers complement the fuzzy number describing a variable by defining its level of certainty or sureness. Since Smets [9] defined perfect information as precise and certain information that is free from inconsistencies, the emergence of Z-numbers can hence be depicted as a new knowledge in the fuzzy set theory that directs into the appreciation of much perfect information. The chronological evolvement of fuzzy sets can be summarized as illustrated in Figure 1.

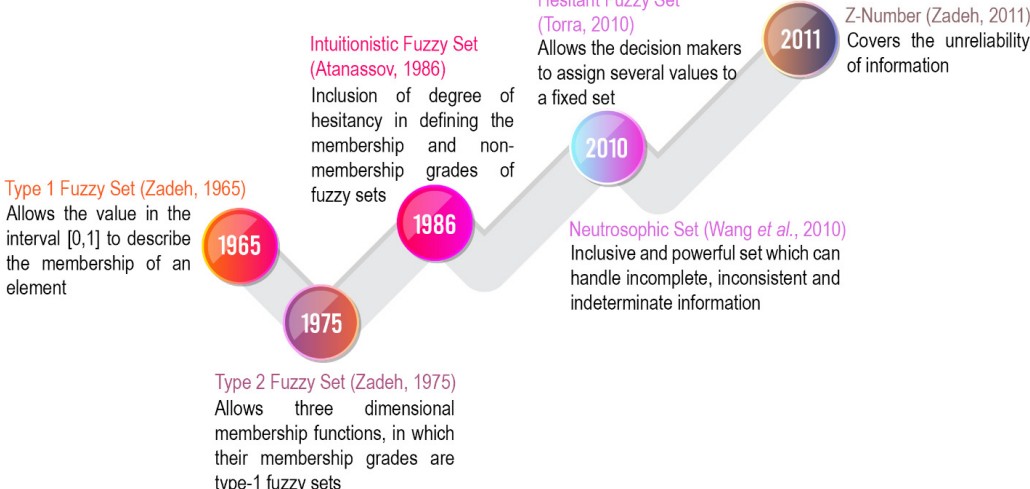

**Figure 1.** The chronology of the extension of fuzzy sets (1965–2011) [1–5,12].

According to Abdullahi et al. [13], a Z-number is a generalization of real, interval, random, and even fuzzy numbers. It was noted that a Z-number has a higher level of generality, which makes it very powerful in realistically modelling real-world systems. The concept of Z-number is capable of better describing imperfect information in expressions

that are almost close to the natural language [14]. This is due to the fact that Z-numbers are capable of representing the uncertainty of the real world as well as the unreliability of human languages [15]. The success of Z-numbers in many applications such as decision making, regression analysis, system control, machine learning, and computing with words has been illustrated in the literature [13].

The research on Z-numbers has attracted many researchers on expanding the knowledge of the extension of fuzzy sets. Based on the Scopus database, there are more than 800 documents whose keyword is Z-number. The number of publications on Z-numbers has shown an increasing trend from 2011 until 2022 with the term 'decision making' leading to the top 10 keywords based on the search of 'Z-number' word in the Scopus database. Among the 10 keywords, the keyword 'decision making' comprises 42.1% (refer to Figure 2). Hence, a refined search on Z-numbers was further performed by combing the keywords 'Z-number' and 'decision making'. The result shows that the number of publications slowly increased from 2012 until 2016. From 2017 onwards, the number of publications tremendously increased, which indicates that the applications of Z-numbers in decision making have been widely studied. Hence, the current research trend on Z-numbers has motivated the authors to conduct a literature review, specifically on the decision making with Z-numbers.

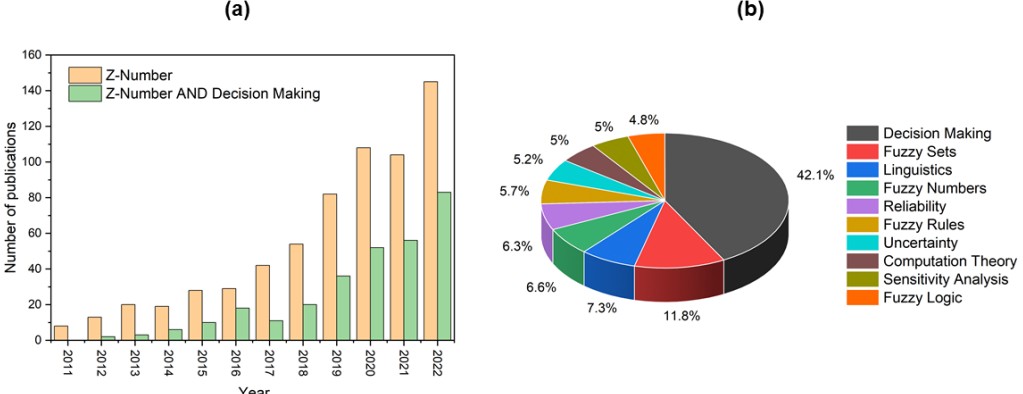

**Figure 2.** (**a**) Number of publications on Z-numbers from the Scopus database (2011–2022). (**b**) Top 10 keywords related to Z-numbers.

There have been several review papers on Z-numbers such as [13,16,17], but the topic is quite wide and not only focused on decision making. The review by [13] focused on the arithmetic concepts of Z-numbers and their applications in decision making, regression, system control, machine learning, and computing with words (CWW). Banerjee et al. [16] reviewed the theoretical concepts and applications of Z-numbers in CWW, decision making, multisensor data fusion, dynamic controller, and safety analytics. Bilgin and Alci [17] specifically reviewed the ranking methods of Z-numbers.

On the other hand, this review paper is more focused on the application of Z-numbers in decision making, with strength, weakness, opportunity, and threat (SWOT) analysis being conducted to evaluate the features of Z-numbers in solving real-world problems. It is believed that each multi-criteria decision-making (MCDM) method has its strength and weakness. Hence, existing MCDM methods based on Z-numbers were explored in this research to make comparative evaluations of the advantages and disadvantages of each of them. Based on the comparative analysis, the strength of each method was identified for the implementation in the development of the MCDM method based on Z-numbers in the future. In fact, the existing MCDM methods are either based on direct computation on Z-numbers or their conversion into regular fuzzy numbers. Both approaches were analyzed to identify the strengths and weaknesses of each approach.

In this review, existing MCDM methods based on Z-numbers are effectively identified. The effectiveness of the hybrid MCDM models that integrate more than one decision-making method was observed. The results of this review can be used to identify the current

state of the MCDM method based on Z-numbers. Hence, researchers in fuzzy decision making are able to find the gap that will expand the knowledge of fuzzy mathematics. Moreover, this review provides a comparative analysis of the ranking approaches of Z-numbers that will help researchers identify the best ranking method to be implemented in the development of the MCDM model using Z-numbers later.

This section provides some background on the emergence of Z-numbers. In Section 2, the review of the theoretical preliminaries of Z-numbers is discussed. Section 3 reviews various decision-making methods and approaches that were developed based on Z-numbers. Section 4 reviews the previously proposed ranking methods of Z-numbers that are closely related to the decision-making concept. Section 5 presents the evaluative analysis of Z-numbers with some important remarks being discussed. Finally, Section 6 concludes this paper.

## 2. Theoretical Preliminaries

The theoretical construction of Z-numbers lies from their conversion into fuzzy numbers, arithmetic operations, and uncertainty measures to the aggregation in the group decision making. Further, the entropy measure of Z-numbers and the method of generating Z-numbers are important concepts in the theoretical construction of Z-numbers. The extension of Z-numbers into many types of other fuzzy Z-numbers has also attracted many researchers in constructing the theoretical background of Z-numbers. These mentioned concepts are summarized and illustrated as shown in Figure 3.

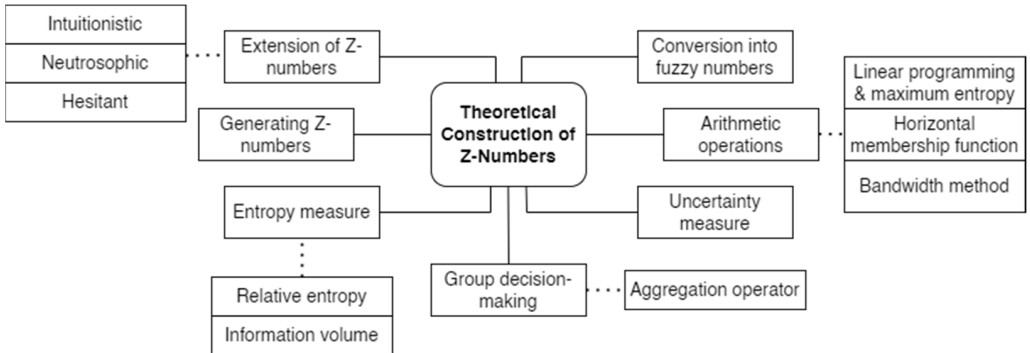

**Figure 3.** The overview of the theoretical construction of Z-numbers.

The calculations involving Z-numbers have a high level of computational complexity since they deal with the possibilistic and probabilistic restrictions [13]. As a way of overcoming this issue, Kang et al. [18] proposed a method of converting Z-numbers into regular fuzzy numbers. In their proposed method, the reliability part of Z-numbers is transformed into a crisp number and treated as a weight, which is then added to the first part of Z-numbers. However, converting Z-numbers into regular fuzzy numbers leads to a significant loss of information [13,19,20]. Later, Aliev et al. [21] proposed arithmetic operations of discrete and continuous Z-numbers, which allow direct calculations on Z-numbers. The proposed operations are mainly regarded with fuzzy arithmetic using $\alpha$-cuts as well as probabilistic arithmetic. These operations do not require the conversion of Z-numbers into regular fuzzy numbers; thus, the issue of information loss can be overcome. Using direct calculations on Z-numbers leads to computational complexity since they require the applications of goal programming extensively just to solve a small problem.

Next, Aliev et al. [22] developed the arithmetic operations of Z-numbers using the horizontal membership functions to overcome the loss of fundamental properties of arithmetic operations over real numbers when the fuzzy arithmetic based on Zadeh's extension principle or the $\alpha$-cuts are used. The proposed approach, which is based on the concept of horizontal membership function and relative distance measure, increases the informativeness of Z-numbers. In real-world applications, the calculations always involve a large number of Z-numbers that need suitable and practical methods in terms of informativeness. Hence, Aliev et al. [23] adopted the bandwidth approach [12] to simplify the computations and the

horizontal membership function [24] for the preservation of the informativeness of Z-numbers. In their approach, the arithmetic between the second components of Z-numbers is treated as a multiplication between fuzzy numbers to reduce the complexity operations over Z-numbers. The obtained value is the approximation of the probabilistic arithmetic and is suitable for applications when a large number of Z-numbers are involved. The representation of Z-numbers as the fuzzy set of probability functions originating from the flexible restriction of the probability of a fuzzy event has led to complex calculations. Hence, Dubois and Prade [25] described the Z-numbers in light of the belief functions and *p*-boxes. The restriction and reliability components of the Z-numbers were treated as crisps, allowing the generated set of probabilities to be representable by a belief function that corresponds to the *p*-box.

An uncertainty measure of Z-numbers was proposed by considering the fuzziness of the constraint and reliability as well as the inherent uncertainty. They also presented certain properties of the developed uncertainty measures such as minimality, maximality, resolution, and symmetry. However, the association between Z-numbers should be compared using the concept of underlying probability distribution [26]. Hence, a new uncertainty measure of discrete Z-numbers is proposed by Li et al. [26] using the maximum entropy method to calculate the underlying probability distributions of Z-numbers. Using the obtained entropies, they formulated a new fuzzy subset in which their basic values are the obtained entropies with the membership functions of the reliability component. However, the uncertainty increases using their proposed uncertainty measure when the reliability component becomes sharper. In fact, when the reliability component becomes sharper, it means that the restriction component becomes more reliable and certain, and hence the uncertainty measure should decrease. Thus, Li et al. [27] modified the uncertainty measure of Z-numbers. In their method, they followed the maximum entropy method in computing the underlying probability distributions similar to [26]. However, the final formula for the uncertainty measure was modified.

Further, Li et al. [28] defined the relative entropy of Z-numbers based on the maximum entropy method and the relative entropy of probability distributions. The maximum entropy was used to calculate the underlying probability distributions of two Z-numbers, $Z_1$ and $Z_2$. The relative entropy between the underlying probability distributions then determined the relative entropy between $Z_1$ and $Z_2$. Recently, Xu and Deng [15] defined the information volume of a Z-number, which can be used to measure the uncertainty level of a Z-number. As the information volume increases, then the uncertainty measure of Z-numbers increases. They used the maximum entropy method to calculate the probability distributions of Z-numbers. They are then transformed into mass functions that are finally integrated using the weighted average.

Some studies have also been conducted to generate Z-numbers from discrete fuzzy numbers. Kang et al. [29] illustrated the case when the fuzzy restriction on a variable is performed with the absence of the reliability part. Hence, they proposed a method of adding the reliability part using the concept of orness measure [30] and maximum entropy. Three situations were considered according to the decision maker's attitude: optimistic, pessimistic, and normative strategies. Tian and Kang [31] modified Kang et al.'s [29] method of generating Z-numbers by generating the ordered weighted averaging (OWA) weights using maximum entropy, generating the value of the reliability part and evaluating its membership function, and measuring the reliability of different decision makers' attitudes using a newly defined similarity measure based on the Hellinger distance. They further illustrated their proposed method using numerical examples, in which the attitudes of decision-makers could be measured based on the reliability values.

For the application of Z-numbers in group decision-making, Aliev et al. [32] defined the Z-number valued *t*-norm and *t*-conorm. The defined *t*-norm and *t*-conorm were then extended to define the aggregation operator to aggregate the decision information in terms of Z-numbers defined by multiple decision-makers. Further, Wang and Mao [33] combined the power average operator [34] with the weighted arithmetic average and weighted geometric average operators for aggregating decision-makers' opinions under the Z-number environment. They successfully applied the defined aggregation operators

in developing a new TOPSIS based on Z-numbers. Peng et al. [35] defined the aggregation operators of Z-numbers based on new operations in light of *t*-norms and *t*-conorms. The Bonferroni mean operator was integrated with the defined operations of Z-numbers to obtain the Z-number-weighted Bonferroni mean. Further, Cheng et al. [36] defined the OWA operator involving Z-numbers. The defined operator was based on Yager's [37] work to take into account the decision maker's attitude in aggregating Z-numbers. Such attitude could be age, experience, knowledge, and other factors [36]. On their defined OWA operator, the hidden probability distribution of Z-numbers was considered, which preserved the Z-number information and highly maintain the original meaning of Z-numbers. In another work by [38], the Bonferroni mean operator was extended to Z-numbers using the overlap functions and grouping functions. They further illustrated the defined operator in solving the new energy investment selection problem under the group decision-making using Z-numbers.

In 2020, Abdullahi et al. [39] proved that the discrete and continuous Z-numbers can be ordered by using a relation between the Z-numbers and any ordered subset of real numbers. The ordered structure of Z-numbers was further used to construct temporal discrete Z-numbers. The supremacy of Z-numbers in describing human knowledge under uncertain environments with limited and partially reliable information has attracted many researchers to extend the knowledge. Sari and Kahraman [40] combined the intuitionistic fuzzy sets with Z-numbers, in which both components of Z-numbers are represented by intuitionistic fuzzy numbers. The inclusion of the membership and non-membership grades of the restriction and reliability components in the intuitionistic Z-numbers (IZN) has improved the ability of Z-numbers in handling uncertainty and partially reliable information. On the other hand, Du et al. [41] combined the neutrosophic fuzzy sets with Z-numbers to produce neutrosophic Z-numbers (NZN). They defined the score function and aggregation operators of NZN and implemented the knowledge in developing a new MCDM method based on NZN. The NZN can express the inconsistency and incompleteness of human judgements since it is capable of describing the truth, falsity and indeterminacy of Z-numbers [42].

## 3. Decision-Making Methods and Approaches

Extensive work has been conducted to help decision-makers in obtaining the best solution to multi-criteria and multi-alternatives problems. The decision-making models can be divided into several groups, which are based on the comparison matrix, distance-based solution, outranking methods, etc. Each of these methods has its strengths and advantages in processing the decision information, as presented in Table 1.

**Table 1.** Prominent MCDM methods with their contributions.

| Reference | Method | Contribution |
|---|---|---|
| [43] | ELECTRE | The ranking of alternatives is done by selecting the best one in which the low-attractive alternatives are eliminated. |
| [44] | DEMATEL | Describes the interrelations among the attributes that can be partitioned into a cause group and an effect group. |
| [45] | AHP | The evaluation of decision-makers is performed using a pairwise comparison matrix. |
| [46] | TOPSIS | The prioritization of alternatives is based on the distance measure from the positive and negative ideal solutions. |
| [47] | PROMETHEE | An outranking method that allows the pairwise comparison of alternatives, in which they are being evaluated according to different criteria, which have to be maximized or minimized. |
| [48] | VIKOR | The ranking and selection from a set of alternatives that allows for the determining of the compromise solutions when there are conflicting criteria. |
| [49] | TODIM | The evaluation of alternatives is based on the dominance degree of each alternative over other alternatives using the overall value. |
| [50] | WASPAS | The ranking of alternatives is determined based on the utility value using the additive and multiplicative relative importance. |
| [51] | CODAS | The best alternative is determined based on the maximum distances from the negative ideal solution. |

In conventional MCDM methods, crisp numbers were used to quantify the level of importance of each criterion and alternative. However, the uncertainty of the decision information was not well simulated using crisp numbers in the traditional MCDM methods. In fact, decision-makers could not express their opinions on alternatives acceptably using crisp numbers due to the complexity of the decision-making scenarios [36]. The implementation of fuzzy sets in decision-making methods was pioneered by Bellman and Zadeh [52]. Since Z-number was first introduced in 2011, the emergence of MCDM methods using Z-numbers hence started to expand from 2012 onwards.

Kang et al. [53] developed a decision-making model using Z-numbers in which the graded mean of triangular fuzzy numbers was used to transform both components of Z-numbers into crisp values and obtain their product. The final priority weight was calculated using the transformed Z-numbers. However, the decision information is not preserved as Z-numbers, which finally leads to a significant loss of information.

The expected utility-based decision-making model using Z-numbers was developed by Zeinalova [54]. The validity of the proposed model was applied to solve the economic problem of investment. However, the proposed model was based on the conversion of Z-numbers. Further, Aliev et al. [55] improved the expected-utility-based model that allows direct calculation on Z-numbers. The Z-numbers were compared using the fuzzy optimality approach.

A TOPSIS model based on Z-numbers was proposed by [56], in which they used the conversion method from [18] to transform Z-numbers into regular fuzzy numbers. Since the decision information was transformed into regular fuzzy numbers, then the distance of each alternative from the ideal solutions was calculated using the distance of fuzzy numbers. They considered the stock selection problem involving 25 companies in Malaysia for the validation of the proposed Z-TOPSIS model.

Babanli and Huseynov [57] employed the arithmetic operations over discrete Z-numbers defined in [58] to solve the optimal alloy selection problem. The fuzzy Pareto optimality was used to make a comparison between Z-numbers, taking into account the degree of pessimism of decision-makers.

Ku Khalif et al. [59] developed the CFPR-TOPSIS method using Z-numbers, in which the criteria weights were evaluated using the CFPR while the TOPSIS was used to rank the alternatives. The intuitive multiple centroid was used to transform Z-numbers into regular fuzzy numbers, in which the reliability component, which is in trapezoidal shape, was partitioned into three parts and the centroid of each sub-area was calculated. Then, the sub-centroids were connected to form a triangle before its center could be located. The *x*-ordinate of the triangle's centroid was used as a weight, which was then added to the restriction component to obtain the regular fuzzy numbers. They validated their proposed model in the selection of the best candidate for staff recruitment in a legal company.

The TOPSIS method was extended into Z-TOPSIS, which was integrated with the principal component analysis and mixed integer linear programming by [60]. However, Kang et al.'s [18] methodology of converting Z-numbers into regular fuzzy numbers was employed in the proposed Z-TOPSIS method. A numerical example of supplier selection in the pharmaceutical industry was used to illustrate the proposed model.

Chatterjee and Kar [61] employed a similar approach to processing Z-numbers in the COPRAS methodology, in which the Z-number was converted into a regular fuzzy number based on Kang et al.'s [18] method to obtain the average subjective weighted value of criteria. The regular fuzzy number was further defuzzified using the centroid formula [62]. Meanwhile, the Shannon entropy, degree of divergence, and objective weight were calculated for the objective part. The crisp values representing both the subjective and objective parts were finally combined using a convex compound to obtain the total weight of each criterion.

Zeinalova [63] developed a novel AHP method based on Z-numbers, in which direct calculation over Z-numbers [58] was used. The fuzzy Pareto optimality was used to

compare between Z-numbers in order to obtain the ranking of alternatives. They applied the proposed Z-AHP in solving the university selection problem.

The Z-VIKOR method was proposed by [19] as an extension of the fuzzy VIKOR method. In the proposed method, the weighted distance measure was proposed, combining the Hellinger distance and the distance of reliability measure. The Hellinger distance was used to quantify the similarity between the underlying probability distributions of two Z-numbers. A numerical example to select the regional circular economy development plan in China was adopted to illustrate the validity of their proposed method.

Krohling et al. [64] developed the TODIM and TOPSIS models based on Z-numbers and validated their models using two numerical examples: the vehicle selection and the clothing evaluation. In both models, the transformation of Z-numbers using the fuzzy expectation [18] was used, which leads to a significant loss of information.

Gardashova [20] presented a TOPSIS method based on the direct calculation of Z-numbers. The arithmetic operations over discrete Z-numbers [58] were implemented to normalize the decision matrix. She used the distance between Z-numbers defined by [65] to calculate the distances between each alternative from the positive and negative ideal solutions. The vehicle selection example [53] was adopted to illustrate the proposed TOPSIS model.

Another TOPSIS model based on Z-number was also proposed by [33]. They defined new Z-number-based arithmetic and geometric aggregation operators to aggregate the decision matrices from multiple decision makers. A new distance measure of Z-numbers was also defined based on the concept of cross-entropy. In their defined distance measure, the first component of Z-numbers was considered to have a higher influence over the reliability component in determining the distance. Supplier selection in the automobile manufacturing industry was considered to validate their Z-TOPSIS model.

In 2020, Tüysüz and Kahraman [66] proposed an extended CODAS methodology based on Z-numbers. The Z-numbers were transformed into regular fuzzy numbers by converting the reliability components into weights using the center of gravity defuzzification, which was then added to the restriction component. The proposed method has low computational complexity involving Z-number calculation, but some information about Z-numbers is dissipated during the transformation into regular fuzzy numbers.

Alternatively, Kang et al. [67] combined the Dempster–Shafer theory (DST) with Z-numbers in estimating the risk of contaminant intrusion in water distribution networks and incidence of infection in hospitals. In their method, they presented a new similarity measure between Z-numbers, which was extended from the utility of fuzzy numbers. Based on the similarities, they determined the basic probability assignment and finally combined it with DST.

An outranking MCDM based on Z-numbers was proposed by [68], namely, the Z-PROMETHEE. They first defined the possible degree of triangular fuzzy numbers. Further, the possibility degrees of both components of Z-numbers, which were represented by triangular fuzzy numbers, were constructed. They were further combined using an adjustable risk preference variable in the convex compound. However, the variable was allocated to take any values in the interval [0,1], which contradicts the fact that the restriction component should be considered as a major factor of Z-number [69]. Their proposed Z-PROMETHEE was successfully applied in solving the travel plan selection problem.

In another study by [70], the linguistic Z-numbers were applied to solve the portfolio selection problem for the stock exchange market in Iran. In their model, the ordered weighted averaging and hybrid weighted averaging aggregation operators were used to aggregate the linguistic Z-numbers. Another approach by [71] used the dynamic programming approach based on Z-numbers to solve the shortest path problem. The shortest path problem is widely applied in transportation and economics [71]. Since no works on solving the shortest path problem based on Z-information, their work was the first to consider the degree of reliability under a fuzzy environment.

Jabbarova and Alizadeh [72] proposed a VIKOR method based on direct computation on discrete Z-numbers. The arithmetic operations from [21] were employed to perform the calculation over Z-numbers in obtaining the regret measures, utility measures, and VIKOR indices. The fuzzy Pareto optimality method was then used to compare between discrete Z-numbers. The proposed VIKOR model was successfully applied in solving the personal selection problem in the recruitment of an online manager in a company.

Aliev et al. [10] proposed an approach to constructing a pairwise comparison matrix whose elements are Z-numbers in a consistency-driven way. They applied differential evolution (DE) to solve the optimization problem with Z-numbers. The method was further applied by [73] in solving the country selection problem.

Many decision-making methods that used the utility function to process Z-numbers were also developed. Ahmadov [74] ranked Z-numbers representing the alternatives using the expected utility function in solving the alternative selection in investment problems. A similar approach was also used by [75] in ranking the alternatives to solve the personal selection problems.

The AHP-WASPAS methodology based on Z-numbers was developed by [76], in which they illustrated the proposed method to prioritize the public services for the implementation of Industry 4.0 tools. In their method, the Z-numbers were transformed into regular fuzzy numbers in both the AHP and WASPAS models. The AHP model was used to obtain the criteria weights, while the WASPAS model was used for the prioritization of the alternatives.

Further, Liu et al. [77] developed the AHP-TOPSIS model based on Z-numbers, in which the Z-AHP was used to obtain the criteria weights while the Z-TOPSIS was to rank the alternatives. In both models, the Z-numbers were converted into regular fuzzy numbers using the fuzzy expectation [18]. The hybrid model was then implemented in the concept design evaluation to select the best waste container in the kitchen.

Another TOPSIS model was developed by [28] using the concept of the relative entropy of Z-numbers. The relative entropy was used to find the separation of each alternative from the ideal solutions, which was further used to obtain the closeness coefficient. They applied the proposed Z-TOPSIS in solving the supplier selection problem. The defined relative entropy measure of Z-numbers allows for direct calculation over Z-numbers; thus, it is able to avoid information loss. However, it leads to high computational complexity, especially for continuous Z-numbers [28]. The DEMATEL was also extended into a Z-number-based methodology by [78]. The extended DEMATEL was then integrated with the HEART method.

Hu and Lin [79] extended the ELECTRE-III into a Z-number-based methodology and weighted Copeland method. In their methodology, the reliability component of the Z-number was transformed into a crisp value using the center of gravity method. The obtained crisp weight was then multiplied by the restriction component to obtain the regular fuzzy number, and the fuzzy entropy method was employed to evaluate the criteria weights. As an illustration, they validated their proposed method for assessing property concealment risk in China.

Another ELECTRE-III method based on Z-numbers was proposed by [80]. They defined the concordance and discordance indices of Z-numbers. Then, the dominance relations of Z-numbers were defined, in which some of the properties were studied. The proposed relations were further applied in the ELECTRE-III model to allow the processing of Z-numbers fully using their bimodal uncertainty so that the nature of Z-numbers could be preserved. Their proposed model was further validated using the renewable energy selection problem [61].

A hybridized VIKOR method with Z-numbers was developed by [81], and they employed supplier selection [60] to validate their proposed model. In their method, the arithmetic mean operator was used to aggregate the preferences from multiple decision makers, in which the restriction and reliability components of Z-numbers were separately aggregated. Both components of Z-numbers were combined using the final rating function

based on the concept of the convex compound, which defuzzified Z-numbers into crisp values to determine the rank of alternatives.

Li et al. [82] defined a generalized distance of Z-numbers to allow for the decision-maker's preferences on different effects on the restriction and reliability parts of Z-numbers. Then, they defined the Z-number gray relational degree for the application in decision-making under the Z-number environment. The effectiveness of their proposed method was illustrated using a Web service selection problem.

In another study by Nuriyev [14], he extended the TOPSIS and PROMETHEE methods into a Z-number-based methodology using the direct calculation of Z-numbers. The proposed models were then applied in the selection of the tourism development sites in Azerbaijan.

A new MCDM method based on Z-numbers and fuzzy Hausdorff distance was proposed by [83] for the selection of the hydro-environmental system to revitalize a lake in Iran. The fuzzy Hausdorff distance was used to avoid the aggregation of Z-numbers using a complicated method and to preserve the initial decision information in the form of Z-numbers.

Jia and Herrera-Viedma [84] used the Pythagorean fuzzy set to solve decision-making problems using Z-numbers. Both components of the Z-numbers were transformed into Pythagorean fuzzy sets, and the Genetic Algorithm was further used to derive the potential probability distribution. The proposed approach was integrated into the decision-making model using linguistic Z-numbers. The model was further applied in solving the energy investment selection problem. The above-mentioned MCDM methods can be further compared and summarized as shown in Table 2.

The TOPSIS based on Z-numbers was observed to be the most developed method among other MCDM models. The developed Z-TOPSIS methods vary in many ways, such as the distance function used to measure each alternative from the ideal solutions, methods of calculation with Z-numbers, and numerical examples as applications. Table 3 lists the previously proposed Z-TOPSIS with their applications.

**Table 2.** Comparative analysis of MCDM methods based on Z-numbers.

| Reference | Method | Processing of Z-Numbers | Ranking of Alternatives | Application | Advantage | Disadvantage |
|---|---|---|---|---|---|---|
| [56] | TOPSIS | Conversion using fuzzy expectation | Euclidean distance between fuzzy numbers | Stock selection problem | Simplified the calculation on Z-numbers | Loss of information |
| [59] | CFPR-TOPSIS | Conversion using intuitive multiple centroid | Euclidean distance from vertical and horizontal centroids | Staff recruitment selection | Improved the method of determining the vertical and horizontal centroids | The reduction of Z-numbers into regular fuzzy numbers does not keep the initial information |
| [60] | PCA-TOPSIS-MILP | Conversion using fuzzy expectation | Distance between fuzzy numbers | Supplier selection in the pharmaceutical supply chain | The integration of PCA reduced the number of criteria | The conversion of Z-numbers leads to a loss of information |
| [61] | COPRAS | Conversion using centroid of triangular fuzzy number | Relative significance and utility degree | Prioritization of renewable energy resources | The combination of subjective weights from decision-makers and objective weights using Shannon entropy | The conversion of reliability parts into centroid leads to the dissipation of information |
| [63] | AHP | Direct calculation on discrete Z-numbers | Pairwise comparison and Pareto optimality principle | Selection of technical institutions | The preservation of information of Z-numbers as no conversion into regular fuzzy numbers was involved | The calculation of hidden probability is tedious |
| [19] | VIKOR | Direct computation of discrete Z-numbers | Hellinger distance of Z-numbers | Selection of regional circular economy development plan | The inclusion of reliability measure and underlying probability distribution in determining the weighted distance of Z-numbers give a more precise measure | Complicated and tedious calculation to solve simple problems |
| [64] | TODIM | Conversion using centroid of trapezoidal fuzzy number | Dominance of alternative over each alternative | Vehicle selection and clothing evaluation | The dominance of alternative over other alternatives is checked one by one | The consideration of centroid of reliability in the conversion dissipates some information |
| [33] | TOPSIS | Paired calculation on restriction and reliability components separately | Weighted paired distance of restriction and reliability components | Supplier selection in an automobile manufacturing company | The weight coefficients of decision makers are obtained via a programming model and the implementation of power aggregation operators in combining the decisions from all decision makers | The final relative closeness coefficient is in pairs of restriction and reliability components, which requires a further approach to combine them |
| [66] | CODAS | Conversion of Z-numbers using center of gravity defuzzification | Euclidean and Taxicab distances of regular fuzzy numbers | Supplier selection problem | The calculation of relative assessment scores based on Euclidean and Taxicab distances | The defuzzification of reliability parts via center of gravity leads to a loss of information |
| [68] | PROMETHEE | Possibility degree of Z-numbers is calculated by combining the restriction and reliability components using a convex compound | Priority index and outgoing and incoming flows | Travel plan selection | The possibility degree of Z-numbers and outranking relations do not involve the conversion of Z-numbers into regular fuzzy numbers | Priority index matrix can only be obtained when the possibility degrees of an alternative over each of the other alternatives are obtained; this is not practical when there are too many alternatives |
| [76] | AHP-WASPAS | Conversion of Z-numbers using center of gravity defuzzification | Utility score combining the weighted sum and product of fuzzy numbers | Prioritization of public services for digitalization | The consideration of weighted sum and product of fuzzy numbers in the utility score can determine the rank of alternatives effectively | The conversion of Z-numbers into regular fuzzy numbers dissipates some information |
| [77] | AHP-TOPSIS | Conversion using fuzzy expectation | Euclidean distance between fuzzy numbers | Conceptual design evaluation of kitchen waste containers | The simplification of the fuzzy TOPSIS method based on Z-numbers based on conversion into regular fuzzy numbers | Loss of information |
| [28] | TOPSIS | The relative entropy of Z-numbers | Relative entropy from the positive and negative ideal solutions | Supplier selection problem | The determination of the underlying probability distributions gives a more precise measure of the reliability components | The consideration of the underlying probability distributions in finding the entropy of Z-numbers made the calculation more tedious |
| [79] | ELECTRE-III | Defuzzification of reliability components into centroid of gravity | Credibility index of fuzzy outranking relation | Property concealment risk ranking | The expert-weight-determining method is introduced in this paper based on group consistency and reliability | The dissipation of information occurs when the reliability components of Z-numbers are defuzzified |
| [80] | ELECTRE-III | Bimodal uncertainty of Z-numbers without conversion into regular fuzzy numbers | Dominance, support, and opposition relations based on Z-numbers | Renewable energy selection problem | The outranking relations based on bimodal uncertainty of Z-numbers have a stronger role in ranking alternatives effectively | The tedious calculation involving the underlying probability of the reliability components despite solving simpler problems |
| [81] | VIKOR | The defuzzification of Z-numbers after obtaining the fuzzy best and worst values | The separation measures from the fuzzy best and worst values | Supplier selection in the pharmaceutical supply chain | The ranking approach based on fuzzy best value and fuzzy worst value can effectively prioritize the alternatives | The defuzzification of the separation measures of the restriction and reliability components dissipates some information before the final ratings are obtained |

**Table 3.** The previously developed Z-TOPSIS models with their applications.

| References | Type of Z-Numbers | Approach | Applications |
|---|---|---|---|
| [56] | Continuous trapezoidal | Conversion of Z-numbers using fuzzy expectation | Selection of stock company |
| [85] | Continuous triangular | Pairwise closeness coefficients of restriction and reliability components | Accident on a construction site by a worker |
| [86] | Continuous triangular | Pairwise closeness coefficients of restriction and reliability components | Selection of agreement from the MoU |
| [60] | Continuous triangular | Conversion of Z-numbers using fuzzy expectation | Supplier selection in a pharmaceutical company |
| [87] | Continuous trapezoidal | Conversion of Z-numbers using intuitive vectorial centroid | Company performance assessment |
| [33] | Continuous triangular | Pairwise closeness coefficients of restriction and reliability components | Supplier selection in the automobile manufacturing industry |
| [64] | Continuous triangular | Conversion of Z-numbers using fuzzy expectation | Vehicle selection [53] and clothing evaluation by male customers [88] |
| [20] | Continuous trapezoidal | Direct calculation on Z-numbers | Vehicle selection [53] |
| [89] | Continuous triangular | Conversion of Z-numbers using fuzzy expectation | Supplier selection in the automobile manufacturing industry |
| [90] | Continuous triangular | Pairwise closeness coefficients of restriction and reliability components | Engineer selection in a software company |
| [91] | Continuous triangular | Choquet integral-based distance | Supplier selection in an enterprise |
| | Continuous triangular | Conversion of Z-numbers using fuzzy expectation | Evaluation of the conceptual design of waste containers |
| [28] | Discrete | Underlying probability distributions and relative entropy of Z-numbers | Supplier selection |

## 4. Ranking Methods

The ranking of fuzzy numbers is a familiar concept to fuzzy MCDM since the final alternatives are prioritized based on the final fuzzy values. In ranking Z-numbers, many methods were developed by researchers in recent years.

The fuzzy Pareto optimality [7] was used by Aliev et al. [55] to rank between discrete Z-numbers. The ranking method calculates three different functions that measure how much a Z-number is better, equivalent, and worse than the other with respect to their both components, which employs the concept of the probability measure. Further, Abu Bakar and Gegov [92] used the vertical and horizontal centroids as well as the spread of fuzzy numbers to rank Z-numbers. In their proposed ranking method, the Z-numbers were first converted into regular fuzzy numbers using Kang et al.'s [18] method. Based on the converted fuzzy numbers, the spreads were calculated, and both the horizontal and vertical centroids were located. Using all these three values, the Z-numbers were ranked. As discussed in the previous section, the conversion of Z-numbers into regular fuzzy numbers leads to a significant loss of information [13,19,20].

Another ranking method was proposed by [93] to determine the ordering of Z-numbers. Similar to [92], the Z-numbers were converted into regular fuzzy numbers before they were standardized and defuzzified. Next, their ranking scores were evaluated, in which the spread of fuzzy numbers was taken into consideration to define the ranking index. On the other hand, Jiang et al. [69] stressed that the reliability component should not be converted into a crisp value to retain as much as possible information contained in Z-numbers. Hence, they proposed another ranking score of Z-numbers based on the centroid point, spread, and Minkowski degree of fuzziness. The scores were separately calculated for both restriction and reliability components. Then, they were combined to determine the final ranking index of Z-numbers, in which the score for the restriction component was given more weight in determining the final ranking index. The underlying support to this construction is

the fact that the restriction component is the main part of Z-numbers while the reliability component is only the peripheral part.

Later in 2018, Kang et al. [94] proposed a utility-based ordering method for Z-numbers, namely, the total utility of Z-numbers. The total utility was an estimation based on the $\alpha$-cuts of the restriction and reliability components of Z-numbers with respect to the interaction of both of them. The derived total utility of Z-numbers was free from subjective membership functions, which was a lacking feature of the defined utility of Z-numbers defined in [21]. Further, Ezadi and Allahviranloo [95] proposed a new method of ranking fuzzy numbers using the hyperbolic tangent function. In their method, the fuzzy number should be normalized before it could be defuzzified using a convex combination. Next, the spread of the fuzzy number was evaluated using the convex combination and standard deviation. The final score of the fuzzy number was defined using the hyperbolic tangent function. The defined score was extended to rank Z-numbers, in which they were converted into regular fuzzy numbers. Then, the similar procedure to rank fuzzy number was employed to rank the converted Z-numbers. The similar approach was also exercised by [96] to rank Z-numbers. The second component of Z-numbers was transformed into a crisp value to be added to the restriction component to obtain regular fuzzy numbers. Then, the sigmoid function and convex combination were employed to rank the fuzzy numbers which were transformed from Z-numbers.

Chutia [97] further used the concept of value and ambiguity at levels of decision-making [98,99] to rank Z-numbers. The value is described as the ill-defined quantities present in a fuzzy number. On the other hand, ambiguity is the level of vagueness contained in the ill-defined quantities [99]. In their proposed method, the values and ambiguities were first calculated before their distances from the origin were found. Combining the values and ambiguities with the evaluated distances, the final value and ambiguity indices were defined. Based on the defined value and ambiguity indices, it can be noted that the restriction component was given higher weightage over the reliability component similar to [69]. For ranking Z-numbers, the value index was given priority. The ambiguity index was only used when the Z-numbers have an equal value index.

The concept of the magnitude of fuzzy numbers was employed by [100] to define the magnitude of Z-numbers in order to obtain their ranking. The magnitude of the restriction and reliability components of Z-numbers was separately evaluated. The magnitude values were then combined using a convex compound, which was used to rank Z-numbers. The use of such a convex compound allowed for the restriction component of Z-numbers to have a higher influence over the reliability component in determining the ranking of Z-numbers. Table 4 summarizes the existing ranking methods based on Z-numbers.

**Table 4.** Summary of the ranking methods of Z-numbers.

| Reference | Method | Limitation |
|---|---|---|
| [55] | Fuzzy Pareto optimality | - |
| [92] | Spread, horizontal centroid, vertical centroid | Conversion of Z-numbers into regular fuzzy numbers |
| [93] | Mean, height, and spread | Conversion of Z-numbers into regular fuzzy numbers |
| [69] | Centroid, spread, and Minkowski degree of fuzziness | - |
| [94] | Total utility of Z-numbers | Involves double conversion from Z-numbers to fuzzy numbers and further converted into crisps |
| [95] | Hyperbolic tangent function and convex combination | Conversion of Z-numbers into regular fuzzy numbers |
| [96] | Sigmoid function and convex combination | Conversion of Z-numbers into regular fuzzy numbers |
| [97] | Value and ambiguity | The ignorance of the ambiguity index when the value index is not unique |
| [100] | Magnitude value | The magnitude could not make a difference on Z-numbers having similar central points with different spreads |

Further, these ranking methods are compared for ranking Z-numbers under different situations, as illustrated in Figure 4. Some ranking approaches are considered for quantifying the Z-numbers.

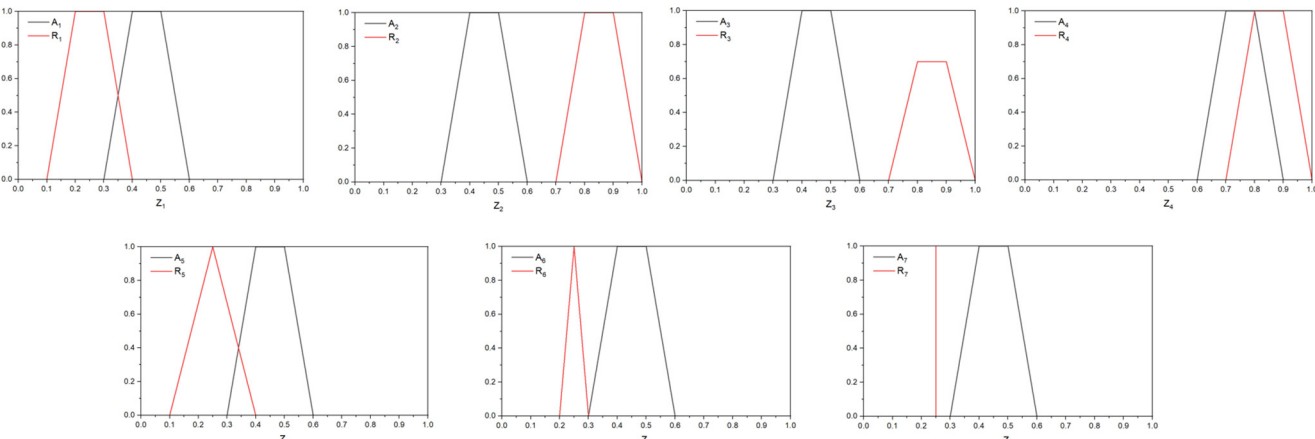

**Figure 4.** Z-numbers under different conditions.

### 4.1. Multiplication of Graded Mean Integration

Kang et al. [53] suggested that the restriction and reliability components of Z-numbers should be defuzzified into graded mean integration representation, as shown in (1) and (2), respectively.

$$P(A) = \frac{1}{6}(a_1 + 4a_2 + a_3) \tag{1}$$

$$P(R) = \frac{1}{6}(r_1 + 4r_2 + r_3) \tag{2}$$

For ranking Z-numbers, the defuzzified values from (1) and (2) are multiplied together to obtain the final ranking result, as shown in (3).

$$P(A \otimes R) = P(A) \times P(R) = \frac{1}{6}(a_1 + 4a_2 + a_3) \times \frac{1}{6}(r_1 + 4r_2 + r_3) \tag{3}$$

### 4.2. Centroid Point and Spread

Abu Bakar and Gegov [92] applied the conversion of Z-numbers into regular fuzzy numbers using the fuzzy expectation, in which the reliability component is converted into a crisp value, as shown in (4).

$$\alpha = \frac{\int x \mu_R(x) dx}{\int \mu_R(x) dx} \tag{4}$$

The crisp value is then added as a weight into the restriction component, which is further transformed into regular fuzzy number, $Z_A = (a_1, a_2, a_3, a_4; h_A)$. Then, the fuzzy number is ranked based on the centroid point and spread, as shown in (5).

$$CPS(Z_A) = x^*_{Z_A} \times y^*_{Z_A} \times (1 - s_{Z_A}) \tag{5}$$

where

$$x^*_{Z_A} = \frac{\int x f(x) dx}{\int f(x) dx}, \tag{6}$$

$$y^*_{Z_A} = \frac{\int \alpha |Z_{A^\alpha}| d\alpha}{\int |Z_{A^\alpha}| d\alpha}, \text{ and} \tag{7}$$

$$s_{Z_A} = y^*_{Z_A} \times |a_4 - a_1|. \tag{8}$$

### 4.3. Centroid, Height, and Spread

Mohamad et al. [93] suggested converting the reliability component of Z-number into a crisp value $\alpha$ using (4). The converted fuzzy number is firstly standardized by diving it with $\kappa = \max(|a_{ij}|, 1)$, as shown in (9).

$$\widetilde{A} = \left(\frac{a_1}{\kappa}, \frac{a_2}{\kappa}, \frac{a_3}{\kappa}; h_{\widetilde{A}}\right) = \left(\widetilde{a}_1, \widetilde{a}_2, \widetilde{a}_3; h_{\widetilde{A}}\right) \tag{9}$$

where $h_{\widetilde{A}} = \alpha$. Then, the ranking of the Z-number is determined using (10).

$$Rank(\widetilde{A}) = \frac{x^*_{\widetilde{A}} \times h_{\widetilde{A}}}{\left(1 - s_{\widetilde{A}}\right)} \tag{10}$$

where

$$x^*_{\widetilde{A}} = \frac{1}{3}\sum_{i=1}^{3} \widetilde{a}_i, \text{ and} \tag{11}$$

$$s_{\widetilde{A}} = \sqrt{\frac{1}{3}\sum_{i=1}^{3}\left(\widetilde{a}_i - x^*_{\widetilde{A}}\right)^2}. \tag{12}$$

### 4.4. Hyperbolic Tangent Function and Convex Combination

Ezadi and Allahviranloo [95] converted the Z-number into a regular fuzzy number and further standardized it using (9). Then, the score of the Z-number was determined using (13).

$$Score(Z) = \frac{e^{\beta} - e^{-\beta}}{e^{\beta} + e^{-\beta}}. \tag{13}$$

where

$$\beta = x^*_{\widetilde{A}} \times |\alpha| + s_{\widetilde{A}} \tag{14}$$

in which

$$x^*_{\widetilde{A}} = \frac{\lambda(\widetilde{a}_2 + \widetilde{a}_3) + (1 - \lambda)(\widetilde{a}_1 + \widetilde{a}_4)}{\lambda\delta_1 + (1 - \lambda)\delta_2}, \; \delta_1 = \delta_2 = 2, \text{ and} \tag{15}$$

$$s^*_{\widetilde{A}} = \sqrt{\frac{\lambda\left(\left(\widetilde{a}_2 - x^*_{\widetilde{A}}\right)^2 + \left(\widetilde{a}_3 - x^*_{\widetilde{A}}\right)^2\right) + (1 - \lambda)\left(\left(\widetilde{a}_1 - x^*_{\widetilde{A}}\right)^2 + \left(\widetilde{a}_4 - x^*_{\widetilde{A}}\right)^2\right)}{(\lambda\delta_1 + (1 - \lambda)\delta_2) - 1}}. \tag{16}$$

### 4.5. Sigmoid Function and Convex Combination

Ezadi et al. [96] used the conversion of Z-number into a regular fuzzy number and further ranked it using the sigmoid function, as shown in (17).

$$Score(Z) = \begin{cases} \frac{1}{1+e^{-\beta}} & , \beta > 0 \\ 0 & , \beta = 0 \\ \frac{-1}{1+e^{-\beta}} & , \beta < 0 \end{cases} \tag{17}$$

where $\beta$ is defined as (14).

### 4.6. Value and Ambiguity

Chutia [97] used the value index to rank the Z-number in which Z-number with a higher value index is ranked higher. The value index of the Z-number $Z = (A, R)$ is defined in (18).

$$I^{val}_{\alpha}(Z) = \frac{V_{\alpha}(A) + \sqrt{2}d^{val}_{\alpha}}{3} \tag{18}$$

where

$$V_\alpha(A) = \frac{1}{2}(a_1 + a_4)\left(h_A^2 - \alpha^2\right) + \frac{1}{3h_A}(a_2 - a_1 - a_4 + a_3)\left(h_A^3 - \alpha^3\right), \text{ and} \qquad (19)$$

$$d_\alpha^{val} = \sqrt{V_\alpha(A)^2 + V_\alpha(R)^2}. \qquad (20)$$

If the value index of two Z-numbers is the same, then the ambiguity index will be used to rank the Z-numbers. The Z-numbers with a lower ambiguity index are more preferred, which is defined as (21).

$$I_\alpha^{amb}(Z) = \frac{A_\alpha(A) + \sqrt{2}d_\alpha^{amb}}{3} \qquad (21)$$

where

$$V_\alpha(A) = \frac{1}{2}(a_4 - a_1)\left(h_A^2 - \alpha^2\right) - \frac{1}{3h_A}(a_2 - a_1 + a_4 - a_3)\left(h_A^3 - \alpha^3\right), \text{ and} \qquad (22)$$

$$d_\alpha^{amb} = \sqrt{A_\alpha(A)^2 + A_\alpha(R)^2}. \qquad (23)$$

### 4.7. Magnitude Value

Farzam et al. [100] defined the magnitude value of Z-numbers for the ranking purpose. The magnitude of the trapezoidal fuzzy number $A = (a_1, a_2, a_3, a_4; h_A)$ was first defined as shown in (24).

$$Mag(A) = \frac{(3h_A^2 + 2)(a_2 + a_3) + (3h_A - 2)(a_1 + a_4)}{12h_A} \qquad (24)$$

Then, the ranking of the Z-number, $Z = (A, R)$, was determined by the convex combination of the magnitude of the restriction and reliability components, as shown in (25).

$$Rank(Z) = \lambda Mag(A) + (1 - \lambda)Mag(R) \qquad (25)$$

where $\lambda \in [0.5, 1]$.

### 4.8. Momentum Ranking Function

The momentum ranking function was proposed by [101] to rank Z-numbers, in which the first component is defuzzified into the center of gravity, as shown in (26), while the second component is defuzzified into the median of fuzzy numbers, as shown in (27).

$$M(A) = \frac{a_3^2 + a_4^2 + a_3 a_4 - a_1^2 - a_2^2 - a_1 a_2}{3(a_3 + a_4 - a_1 - a_2)} \qquad (26)$$

$$M(R) = \frac{a_1 + a_2 + a_3 + a_4}{4} \qquad (27)$$

Hence, the final ranking of the Z-number is determined using the momentum ranking function defined in (28).

$$Rank(Z) = M(A, R)(Z) = M(A) \times M(R) \qquad (28)$$

### 4.9. Comparison of Ranking Methods

Comparing $Z_1$ and $Z_2$, all the ranking methods ranked $Z_2$ higher than $Z_1$, except for [92]. In reference to Figure 4, $Z_2$ should be ranked higher than $Z_1$ since its reliability component is much larger. When $Z_2$ is compared to $Z_3$, $Z_2$ should be ranked higher because its reliability component has a higher maximum membership value. However, all methods could not validate this result, except for [97,100]. This is because both the ranking

approaches considered the height of fuzzy numbers in determining the final ranking of Z-numbers.

For comparing $Z_1$ and $Z_4$, both the restriction and reliability components are considered since they do not share the same fuzzy numbers at all. In this case, $Z_4$ should be ranked higher due to the fact that it has larger fuzzy numbers representing both the restriction and reliability components. This situation is successfully validated using all the ranking approaches without any exceptions.

Next, the effect of triangular and trapezoidal shapes of the reliability component sharing the same spread were validated by comparing $Z_1$ to $Z_5$. All the ranking methods ranked $Z_1$ similar to $Z_5$, except for [97]. Since the value indices of both $Z_1$ and $Z_5$ are the same as presented in Table 5, Chutia [97] suggested using the ambiguity indices to compare the Z-numbers. In this case, $I_\alpha^{amb}(Z_1) = 0.0569$ and $I_\alpha^{amb}(Z_2) = 0.0670$. Hence, $Z_1$ is ranked higher than $Z_2$ since its ambiguity index is much smaller. Among all the ranking methods observed, only the value and ambiguity approach could make a difference between the triangular and trapezoidal fuzzy numbers representing the reliability component.

Next, $Z_5$ and $Z_6$ were compared to see the effect of different spreads of the triangular fuzzy numbers representing the reliability components. Again, only Chutia [97] managed to make a difference in the ranking of Z-numbers using the ambiguity index. For this situation, $Z_6$ has a lower ambiguity index, which is 0.0295, compared to $Z_5$ with a 0.0670 value for the ambiguity index. Hence, the rule by Chutia [97] concluded that $Z_6$ is ranked higher. This result is further supported by the fact that fuzzy number with smaller spread is more preferred [102].

Next, a singleton representing the reliability component of Z-number was compared to the triangular fuzzy number by the illustration of the comparison between $Z_6$ and $Z_7$, as shown in Figure 4. Again, all ranking methods were unable to make a difference between the ranking of $Z_6$ and $Z_7$, except for [97]. Since the value indices of both $Z_6$ and $Z_7$ share the same value, the ambiguity index was hence used to make the comparison. Using (21), the ambiguity indices of $Z_6$ and $Z_7$ were obtained as 0.0295 and 0.0236, respectively. Therefore, it could be concluded that $Z_7$ is ranked higher than $Z_6$ since it has a smaller ambiguity index. This result was due to the fact the singleton does not possess the fuzziness property.

Therefore, the ranking approach based on the value and ambiguity is most suitable for ranking the Z-numbers. This is based on its ability in making comparisons on the triangular and trapezoidal shapes, different spreads, and different maximum membership values. However, other ranking approaches such as magnitude value [100] could also be used with the improvement on the consideration of the spread in defining the magnitude of Z-numbers.

**Table 5.** Ranking results of Z-numbers using various approaches.

| Z-Number | | | Ranking Approaches | | | | | | | |
|---|---|---|---|---|---|---|---|---|---|---|
| **Z** | **A** | **R** | **[53]** | **[92]** | **[93]** | **[95]** | **[96]** | **[97]** | **[100]** | **[101]** |
| $Z_1$ | (0.3,0.4,0.5,0.6;1) | (0.1,0.2,0.3,0.4;1) | 0.1125 | 0.1641 | 0.1012 | 0.2642 | 0.5672 | 0.2945 | 0.3500 | 0.1125 |
| $Z_2$ | (0.3,0.4,0.5,0.6;1) | (0.7,0.8,0.9,1.0;1) | 0.3825 | 0.1641 | 0.3440 | 0.4935 | 0.6320 | 0.4525 | 0.6500 | 0.3825 |
| $Z_3$ | (0.3,0.4,0.5,0.6;1) | (0.7,0.8,0.9,1.0;0.7) | 0.3825 | 0.1641 | 0.3440 | 0.4935 | 0.6320 | 0.2984 | 0.5863 | 0.3825 |
| $Z_4$ | (0.6,0.7,0.8,0.9;1) | (0.7,0.8,0.9,1.0;1) | 0.6375 | 0.2734 | 0.4829 | 0.6616 | 0.6890 | 0.5883 | 0.8000 | 0.6375 |
| $Z_5$ | (0.3,0.4,0.5,0.6;1) | (0.1,0.25,0.25,0.4;1) | 0.1125 | 0.1641 | 0.1012 | 0.2642 | 0.5672 | 0.2945 | 0.3500 | 0.1125 |
| $Z_6$ | (0.3,0.4,0.5,0.6;1) | (0.2,0.25,0.25,0.3;1) | 0.1125 | 0.1641 | 0.1012 | 0.2642 | 0.5672 | 0.2945 | 0.3500 | 0.1125 |
| $Z_7$ | (0.3,0.4,0.5,0.6;1) | (0.25,0.25,0.25,0.25;1) | 0.1125 | 0.1641 | 0.1012 | 0.2642 | 0.5672 | 0.2945 | 0.3500 | 0.1125 |

## 5. Evaluative Analysis

The SWOT in Figure 5 describes the strengths and weaknesses of Z-numbers, which comes along with the opportunities and threats to Z-numbers. The discussed features of Z-numbers in this section are more focused on their applications in decision making.

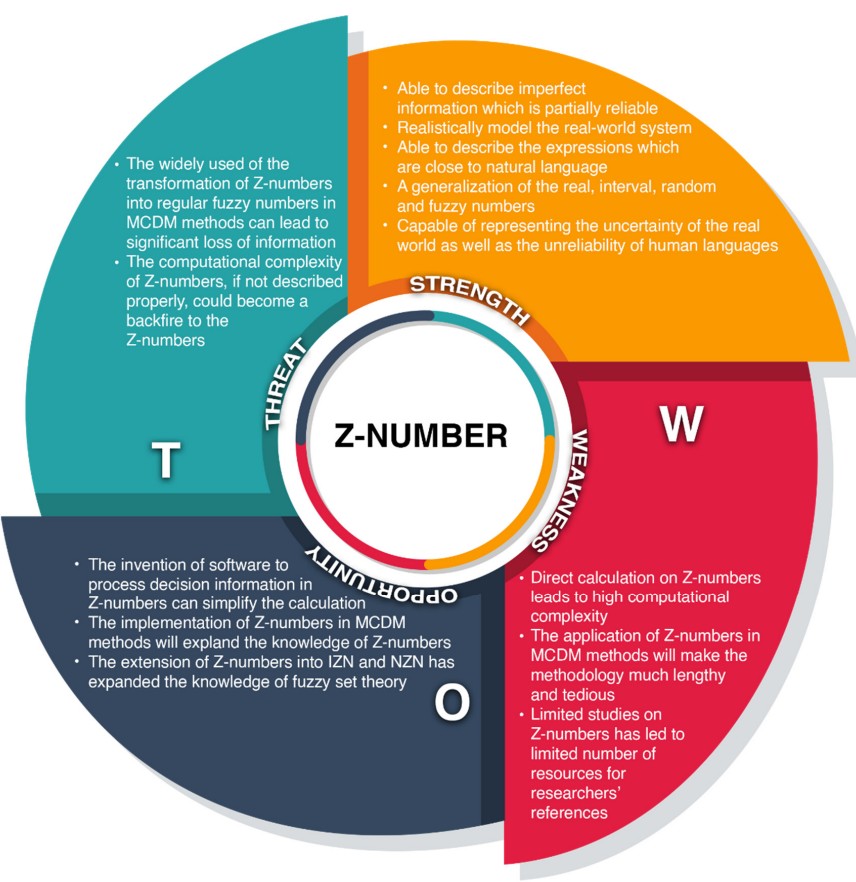

**Figure 5.** Strength, weakness, opportunity, and threat of Z-number.

Based on the SWOT analysis of the application of Z-numbers in decision making, some key findings are mentioned as follows:

- The hybridization of more than one MCDM method using Z-numbers produced a better result [59]. When applied to MCDM models, the selection of alternatives is much better than the single MCDM model based on Z-numbers. In fact, the hybrid MCDM methods are designed to cancel out the drawbacks of their respective methods when used alone [103].
- The invention of software that can process decision information in the form of Z-numbers is vital to simplify mathematical calculation [14]. The availability of such software helps experts from other fields such as business, economy, finance, psychology, and education solve their problems that involve various attributes and alternatives.
- When ranking Z-numbers, the first component should be given at least a higher weightage than the second component [69,100]. This is supported by the fact that the restriction component is the main part, while the reliability component is a subordinate part to Z-numbers.
- It is important to note that Z-numbers are not only composed of the restriction and reliability components, but the hidden probability distribution is another important concept regarding Z-numbers since it connects the restriction component to the reliability component [36].

## 6. Conclusions

The theory of Z-numbers has been widely applied in developing various MCDM models, which have become an interesting topic in fuzzy mathematics. Among all Z-number-based MCDM methods, TOPSIS was found to be the method that was most developed with various applications. The MCDM methods based on Z-numbers appeared to have two major ways of processing Z-numbers: conversion into regular fuzzy numbers and direct computation over Z-numbers. The first scenario seems to cause a significant loss of information since the initial information in the form of bimodal uncertainty is damaged. However, direct computation over Z-numbers leads to high computational complexity. Hence, more references and innovative software should be made available for the public to understand the information processing using Z-numbers such that the knowledge of this fuzzy set theory could be appreciated. For the application of Z-numbers in MCDM, the hybridization of more than one method has produced a more consistent result compared to the single model. Hence, for developing MCDM models based on Z-numbers in the future, at least two models should be considered. Each MCDM model should be critically analyzed as each model has its strengths and advantages in computing the criteria weights as well as producing consistent alternative ranking. The ranking function is also another important concept related to the application of Z-numbers in decision making. It is important to note that the first component of Z-numbers should be given a higher weightage when developing any ranking method of Z-numbers. Hence, implementing the best ranking function for Z-numbers promises a more consistent result in solving any decision-making problems that are associated with information reliability.

However, this research is limited to the application of Z-numbers in decision making. Z-numbers have been extended to intuitionistic Z-numbers, neutrosophic Z-numbers, Pythagorean Z-numbers, Fermatean Z-numbers, and Q-rung orthopair Z-numbers. These generalized Z-numbers are a better representation of the Z-numbers, which have advantages worth studying. Hence, future research on the Z-numbers should expand the MCDM models based on these generalized Z-numbers, and their strengths should be effectively identified. In addition, a method of considering the bimodal uncertainty of the Z-numbers by considering the underlying probability distributions contained in the Z-numbers should be developed based on a simpler calculation approach. This is important in order to make sure that the implementation of Z-numbers is practical in designing decision-making models. Hence, they can be widely used by people without fuzzy mathematics background to solve real-world problems. This action will definitely be a huge contribution to the worldwide community with the appreciation of fuzzy mathematics knowledge.

**Author Contributions:** Main text, N.M.F.H.N.B.A. and K.M.N.K.K.; SWOT analysis, N.I.J. and A.G.; supervision, K.M.N.K.K. and A.G.; funding acquisition, K.M.N.K.K. All authors have read and agreed to the published version of the manuscript.

**Funding:** This research was funded by Universiti Malaysia Pahang under UMP Postgraduate Research Grants Scheme (PGRS) No. PGRS220301.

**Acknowledgments:** The authors would like to thank Universiti Malaysia Pahang for laboratory facilities as well as additional financial support under the UMP Postgraduate Research Grants Scheme (PGRS) No. PGRS220301.

**Conflicts of Interest:** The authors declare no conflict of interest.

## Abbreviations

| | |
|---|---|
| AHP | Analytic hierarchy process |
| CFPR | Consistent fuzzy preference relations |
| CODAS | Combinative distance-based assessment |
| COPRAS | Complex proportional assessment |
| CWW | Computing With Words |
| DE | Differential evolution |

| | |
|---|---|
| DEMATEL | Decision making trial and evaluation laboratory |
| DST | Dempster–Shafer theory |
| ELECTRE | Élimination et choix traduisant la réalité |
| GA | Genetic algorithm |
| HEART | Human error assessment and reduction technique |
| IZN | Intuitionistic Z-number |
| MCDM | Multi-criteria decision making |
| MILP | Mixed integer linear programming |
| NL | Natural language |
| NZN | Neutrosophic Z-number |
| OWA | Ordered weighted averaging |
| PCA | Principle component analysis |
| PROMETHEE | Preference ranking for organization method for enrichment evaluation |
| SWOT | Strength, weakness, opportunity, and threat |
| TODIM | Tomada de decisao interativa multicriterio |
| TOPSIS | Technique for order of preferences by similarity to ideal solutions |
| VIKOR | Visekriterijumska optimizacija I kompromisno resenje |
| WASPAS | Weighted aggregated sum product assessment |

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
