# Peer review of "The Application of Z-Numbers in Fuzzy Decision Making: The State of the Art"

_information, doi:10.3390/info14070400_

Round 1

Reviewer 1 Report

This paper makes a study of the state of the art of the Z-numbers theory.

The paper is well structured and it is well written although English should be revised because there are some mistakes, for instance, “their respective methods when used singularly

In the introduction, the acronym NL is explained, however the acronyms SWOT and CWW are not explained. Although there is a table with all acronyms used in the per, all of them should be explained when tehy appear the first time.

SEction 3, “the best solution of multi-criteria and multi-alternatives problema..” should be “the best solu-247 tion of multi-criteria and multi-alternatives problems…”

Section 3, “In the proposed method, The weighted…” should be “In the proposed method, the weighted…”

To sum up, the paper make a good reviews of the Z-Numbers applied to decision making because it has been detected that there are many recent publications on such a application.

English should be revised because there are some minor mistakes. For example, Section 3, “the best solution of multi-criteria and multi-alternatives problema..” should be “the best solu-247 tion of multi-criteria and multi-alternatives problems…”

Reviewer 2 Report

In this manuscript, the authors review the existing similarity measures, ranking methods and decision-making models based on Z-numbers.

My remarks are as follows:

1. The motivation of this paper should be further strengthened and the main contributions should be pointed out.

2. Section “3. Decision-making Methods and Approaches” is too narrative. Here, an analytical part should be added. The authors should summarize the key features of existing approaches and provide a comparative analysis of the results obtained.

3. In the “3. Ranking Methods” section, an analytical segment needs to be included. This should address the advantages and disadvantages of existing techniques for Z numbers ranking.

4. The “5. Conclusion” section should be extended. Study limitations and future plans should be added.

Technical remarks:

Please, edit the text to avoid some repetitions, for example:

l. 20-21: “reliable”, “reliability”;

l. 24-25: “reviews”, “reviewed”, etc.

Figure 1 is not mentioned in the text.

Figure 2 duplicates a figure that was previously published by other authors in a previous study and could be omitted.

Table 3 in not mentioned in the text.

Minor editing of English language is required.

Round 2

Reviewer 2 Report

The quality of information-2480857-peer-review-v2 “The application of Z-numbers in fuzzy decision-making: A state-of-the-art” has been considerably improved.

In my opinion, the manuscript meets the requirements of the Information Journal.

My recommendation is “Accept as is”.

Minor editing of English language is required.